# Generating Road Networks for Old Downtown Areas Based on Crowd-Sourced Vehicle Trajectories

**DOI:** 10.3390/s21010235

**Published:** 2021-01-01

**Authors:** Caili Zhang, Yali Li, Longgang Xiang, Fengwei Jiao, Chenhao Wu, Siyu Li

**Affiliations:** 1State Key Laboratory of Information Engineering in Surveying, Mapping and Remote Sensing, Wuhan University, Luoyu Road 129, Wuhan 430079, China; cailizhang@whu.edu.cn (C.Z.); liyali@whu.edu.cn (Y.L.); fwjiao@whu.edu.cn (F.J.); ngch@whu.edu.cn (C.W.); 2Urban and Rural Construction College, Shaoyang University, Xueyuan Road, Daxiang District, Shaoyang 422000, China; 3School of Resource and Environmental Sciences, Wuhan University, Luoyu Road 129, Wuhan 430079, China; lsrain@whu.edu.cn

**Keywords:** crowd-sourced vehicle trajectories, old downtown areas, intersection extraction, link identification, Delaunay triangulation network

## Abstract

With the popularity of portable positioning devices, crowd-sourced trajectory data have attracted widespread attention, and led to many research breakthroughs in the field of road network extraction. However, it is still a challenging task to detect the road networks of old downtown areas with complex network layouts from high noise, low frequency, and uneven distribution trajectories. Therefore, this paper focuses on the old downtown area and provides a novel intersection-first approach to generate road networks based on low quality, crowd-sourced vehicle trajectories. For intersection detection, virtual representative points with distance constraints are detected, and the clustering by fast search and find of density peaks (CFDP) algorithm is introduced to overcome low frequency features of trajectories, and improve the positioning accuracy of intersections. For link extraction, an identification strategy based on the Delaunay triangulation network is developed to quickly filter out false links between large-scale intersections. In order to alleviate the curse of sparse and uneven data distribution, an adaptive link-fitting scheme, considering feature differences, is further designed to derive link centerlines. The experiment results show that the method proposed in this paper preforms remarkably better in both intersection detection and road network generation for old downtown areas.

## 1. Introduction

Road networks are of great significance to urban development and for traveling. How to obtain road information for reasonable planning and resource allocation has always been an economic issue for national economies and people’s livelihoods [1]. With the development of surveying, mapping, communications, computers, and other technologies, we can infer road networks based on various data sources, such as crowd-sourced vehicle trajectories [2,3,4,5], laser point clouds [6,7], remote sensing images [8,9], aerial images [10,11,12], OpenStreetMap [13,14,15], etc. Among these data sources, crowd-sourced trajectories have become mainstream data sources of generating road information, and have triggered a large amount of research on road extraction in the past few years, focusing on prominent features, such as wide coverage, high update frequency, and low acquisition cost [16].

However, some challenges still exist in extracting road elements for old downtown areas, based on crowd-sourced vehicle trajectories. On the one hand, old downtown areas has a complicated road network structure, making it the most difficult area for road network extraction, which is mainly reflected in the following two aspects:
The distance between road intersections/road segments is narrow (Figure 1b). Compared with other regions, old downtown areas has high-density buildings and people. In order to ensure good traffic capacity, old downtown areas has been renovated many times and the roads are much denser.The road network of old downtown areas is mixed with primary and secondary roads. The main roads in old downtown areas form the basic road network frameworks, with branch roads scattered throughout. However, other areas (Figure 1b) are still under development, and the roads are relatively wide, with little difference in road grades.

On the other hand, the quality of the vehicle trajectories in the old downtown areas is relatively low and the characteristics of road networks in old downtown areas form unique trajectory distributions, which affect the effective extraction of road networks. It can be reflected in the following three aspects:
The low accuracy of the vehicle receiving equipment and interference from road surroundings to Global Positioning System GPS) signals has caused serious noise for crowd-sourced vehicle trajectories [17], which induce spatial uncertainties and increase the difficulty of knowledge mining (Figure 1a).Crowd-sourced vehicle trajectories are usually sparsely sampled (the red track in Figure 1a), and the trajectories of some roads are densely distributed [18]. Therefore, adjacent intersections or road segments are difficult to distinguish.The mixture of trunk roads and branches in old downtown areas directly leads to the over concentration of traffic flow on the main roads and fewer trajectory points on the secondary road [19], which increases the difficulty of extracting the complete road network (yellow district in Figure 1a).

Due to the challenges above, the existing methods do not work well when using crowd-source trajectories to extract roads in old downtown areas. Cao and Krumm [20] cannot effectively fuse road segments to form the road network, while, Edelkamp and Schrödl [21] can only detect cluster points, as shown in Figure 2a,b. The intersection linking method proposed by Karagiorgou and Pfoser [22] also does not work well and takes more than 1 week to generate results based on our experimental data. Even if the raster method proposed by Davies [23] can form the road network, the adjacent road segments in old downtown areas cannot be effectively distinguished, and road segments around the intersections are deformed, as shown in Figure 2c. Furthermore, raster methods produce many burrs and affect the connectivity of the road network.

To this end, this paper adopts a novel intersection priority strategy to address the aforementioned challenges to automatically generate a road network of old downtown areas, based on crowd-sourced big trajectory data. First, intersections are extracted by clustering virtual representative points, and then the different category link fitting methods are used to infer road segments based on the guidance information of intersections, so as to construct the road network of old downtown areas in a divide and conquer manner. The main contributions are as follows:
Virtual representative points, considering distance constraints, were designed to eliminate the influence of curve segments and noise points. On this basis, the clustering by fast search and find of density peaks (CFDP) algorithm was introduced to detect intersections, which overcomes the sparseness of trajectory sampling and ensures the accuracy of intersection positioning.A corresponding strategy of links identification based on the Delaunay triangulation network was established according to characteristics of road structure and trajectory distribution, which avoids the calculation of redundant links and guarantees the generation of more realistic structures.An adaptive link-fitting scheme, considering feature differences, was designed to effectively alleviate the curse of sparse and uneven distribution and ensure the precision of the extraction results. In addition, a new method based on piece-wise link fitting, focusing on sparse GPS road segments, was proposed.

## 2. Related Work

The extraction of road network for old downtown areas is of great significance and directly affects the quality of urban construction and development. However, the complex structure of road network in old downtown areas and the low quality of the crowd-sourced trajectories have brought a series of challenges for road network extraction [24]. Therefore, it is necessary to make a great contribution to extracting geometric (or attribute information) in old downtown areas, based on crowd-sourced trajectories, automatically.

At present, an inferring road network based on crowd-sourced trajectory data is a hot spot, and some researchers have completed several seminal works, which can be divided into incremental methods [25,26,27], clustering methods [28,29,30], raster methods [31,32], and intersection-link methods [33,34,35]. Incremental methods conform to the law of human cognition, continuously add new trajectory lines to merge with the previous generated lines to form a road network, but cannot optimize the abnormal trajectory of low-frequency trajectory data, and are sensitive to noise [36]. Clustering methods mainly detect road feature points or clusters to infer road networks. Raster methods extract road centerlines by processing the raster image converted from the original GPS trajectories. These two methods can effectively solve the low frequency problem, but cannot distinguish two roads that are close in space. In addition, the three methods above cannot guarantee the position of road intersections leading to generation of many unrealistic structures that are distorted near intersections, and cannot infer the road segments in low-grade roads or sub-district roads with sparse trajectories [37]. In sum, the three methods are not available for old downtown areas to extract road networks directly based on crowd-sourced trajectories.

Intersection-link methods detect road intersections first based on density distribution of trajectory sampling points and their implicit semantic features [15,38], trajectory point direction, speed, and their implicit dynamic features [17,39], and then connect these intersections to form the road network. However, current research mainly focuses on intersection extraction, and seldom conduct further road network generation [40]. Moreover, most road generation methods are based on high-frequency trajectories [41,42]. Recently, a challenge piqued the interest of some researchers, and several new solutions were proposed [43,44,45] to calculate the road segments. However, this challenge was also based on a high-quality trajectory date. Thus, intersection-link methods mentioned above are also not available for road network generation of old downtown areas from crowd-sourced trajectory data.

In our previous work, we designed intersection-priority urban road network generation technology from crowd-sourced trajectory data, which combines mathematical morphology processing and CFDP. However, the features for intersection and road extraction are more suitable for dense areas. Considering the importance of road intersections, a more effective method for road network generation based on intersection extraction results, which consider low-frequency characteristic of GPS traces and the knowledge of old downtown areas road network surroundings, has been developed.

## 3. Road Network Generation Method

Due to the road characteristics of old downtown areas, in order to ensure that the road extraction results near intersections are not distorted, we adopt an intersection-link scheme to infer the road network of old downtown areas based on the analysis above. Unlike other approaches of calculating links directly after intersection detection, our method first identifies links and then creates road segments, which can make road extraction faster and more precise. The corresponding road information extraction scheme for old downtown areas, including three key parts, as shown in Figure 3, are:
Road intersection extraction. In order to obtain more accurate road intersections, we extracted representative points by limiting the distance of turning point pairs, then performed Kernel Density Estimation (KDE) for data smoothing, and finally extracted the road intersections by the CFDP algorithm.Link identification. Delaunay triangulation network was constructed, and corresponding judgment criteria were proposed to identify links based on trajectory distribution and road structure features. We also fused the road extraction results based on the morphology method [1] to optimize true link identification.Targeted link fitting. Based on the above process, for different types of links, three different fitting methods were used to infer road segments. Straight line fitting and optimizing result fitting were used for dense GPS road segments, and a new piece-wise fitting method was proposed for sparse GPS road segments to effectively alleviate the curse of trajectory data sparse and uneven distribution, which can ensure the integrity of the extraction results.

### 3.1. Intersection Detection Based on CFDP with Representative Points

Road intersections play a significant role in the road network connection. Inspired by the phenomenon that vehicle-heading directions will change directly (more than 45°) when a turn process is completed at the road intersections, Wu [46] extracts converging points (intersection points of turning point vectors, as shown in Figure 4a,b) and detects road intersections based on improved X-means algorithm. However, this algorithm requires more parameters, and the intersection positioning accuracy is not high. Furthermore, with the car moving in the curve roads, turning point pairs with long distance will yield many converging points away from the road, which will seriously affect true location detection of road intersections (green points), as shown in Figure 4b. Thus, distance of turning point pairs can be limited and eliminate the influence of curved road sections. The road intersection results (green points) of distance limited are better than non-distance limited as shown in Figure 4c. Distance limit threshold can be set to 200 m, which has a higher frequency in the distance statistics of turning point pairs (Figure 4d).

According to Figure 4c, except concentrated points, there are some discrete points (noise points) distributing around road intersections, which may also result in the detection results deviating their true locations. Therefore, KDE was used for data smoothing, as shown in Figure 4e. Setting appropriate threshold *K* to extract high-density cells and detecting road intersections by CFDP algorithm can guarantee the location precision of road intersections again. The Kernel density estimator at point *x* can be shown in Equation (1):(1)$$\overset{\wedge }{f(x)}=\frac{3}{mh^{2}}\sum_{i=1}^{m}K(\frac{1}{h}(x-x_{i}))$$
where *m* is the number of neighbor cells, *x**_i_* is the center point of the *i*-th cell, *h* is the bandwidth, and *K*(*x*) is the kernel function adopted in this work, as shown in Equation (2):(2)$$K(x)=\{\begin{array}{c} 3\pi^{-1}{\left(1-X^{T}X\right)}^{2},X^{T}X<1\\ 0,otherwise \end{array}$$

CFDP algorithm is used to detect road intersections thanks to its threshold settings and stability of results [1]. In order to find density peaks, this algorithm needs to calculate the local density and distance of cell points. Due to estimating density processing, high-density cells have had the density attributes ${\left\{\rho_{i}\right\}}_{i=1}^{N}$ and their distance attributes $\delta_{qi}$ can be calculated based on Equation (3):(3)$$\delta_{qi}=\{\begin{array}{l} \underset{\begin{array}{l} qj\\ j<i \end{array}}{\min}\{d_{qiqj}\},i\geq 2\\ \underset{j\geq 2}{\max}(\delta_{qj}),i=1 \end{array}$$
where ${\left\{q_{i}\right\}}_{i=1}^{N}$ is the descending order of ${\left\{\rho_{i}\right\}}_{i=1}^{N}$.

Setting appropriate distance threshold *d* and omitting the density threshold can obtain more road intersections, which not only locate in high-density areas, but also low-density areas. Therefore, according to the decision graph (Figure 4f); threshold *d* can be set to 20 m.

It must be mentioned that after the processing above, some false intersections still exist in the extraction results. Pseudo intersections that fall outside the road will affect subsequent extraction of road segments. Therefore, we collect trajectory points that fall into the buffer of radius r_1_ and deleted those results whose count is less than the given threshold c to eliminate the impact of this kind of false intersections. Other false intersections that land on the road and have two or fewer connected roads can be pruned based on the following road network generation results.

For road intersection extraction, the experimental parameters include kernel density threshold *K*, the bandwidth *h*, the cell size *s*, clustering distance threshold *d*, radius *r*_1_ and point number threshold *c*. The parameters of *h*, *s*, *d*, and *r*_1_ are easy to set. In order to distinguish adjacent intersections, bandwidth *h* can be set as the minimum distance between intersections in old downtown areas. Cell size *s* was the minimum width and height of the study area divided by 250. Distance threshold *d* can be easy set according to the decision graph. The parameter *r*_1_ is usually set as the minimum width of road in the study area. The parameters *K* and *c* are set empirically. By default, *c* is set to 10 and *K* is set to one fifth of the average density. These two parameters are difficult to set and require further research.

### 3.2. Link Extraction Based on Delaunay Triangulation Network

After intersection detection, we can connect them to generate road network. For low frequency of crowd-sourced trajectory data and narrow spacing between road segments of old downtown areas, directly traversing the trajectory data to connect road intersections will produce a large number of invalid sections and increase the calculation amount. As a method of constructing topological relationship of data set, the Delaunay triangulation network can reflect the similarity relationship between data objects well, and some links of it are completely consistent with most road links. Therefore, we can first identify which intersections have links based on the adjacency relationship of the Delaunay triangulation network and some hidden rules, and then create road segments. However, some of these links are located in dense trajectory areas, and some are located in sparse trajectory areas. Moreover, some other links in density area cannot be constructed based on the Delaunay triangulation network. Hence, for the above three type links, a targeted road centerline identification, and fitting strategy considering feature differences was designed to infer large-scale road segment by divide-and-conquer calculations. Compared with processing all links formed by permutation and combination of intersections, our proposed method can reduce redundancy and guarantee precise results.

#### 3.2.1. Link Identification

Directly using the Delaunay triangulation network constructed by intersection results for road true links identification will produce pseudo results around the edge of the study area, which is mainly caused by peripheral long and narrow triangles. Therefore, we detected triangles with two common sides, and then deleted those triangles that the angle of two common sides is larger than a certain threshold *T* (by default, *T* = 135°) through iteration to construct the initial links identification network, as shown in Figure 5a. Based on the above processing results, we first give the three type links definition, and then introduce the corresponding identification schemes. The three type links are defined as follows.

Type I links: some links that can be constructed based on the Delaunay triangulation network. They are located in dense trajectory areas and can represent the road links.

Type II links: some links that cannot be constructed based on the Delaunay triangulation network. They are located in dense trajectory areas and can represent the road links.

Type III links: some links that can be constructed based on the Delaunay triangulation network. They are located in sparse trajectory areas and can represent the road links.
Type I links identification: according to observations, we found that true links often contain more trajectory points around them compared with pseudo links. Therefore, we proposed Criterion 1 to help identify true links. However, due to the dense roads in old down town and the small distance between roads, some false links will be identified. Therefore, considering road structure features that urban roads are generally designed to be square and rarely involve triangular forms, and the minimum reference intersection angle of two roads is generally set to 60° [34], Criterion 2 and Criterion 3 were proposed to eliminate false links from candidate true links obtained by Criterion 1. The specific criteria are set as follows:
Criterion 1: assuming that Tr is the trajectory data set, L is the triangle edge. We divided L into *m* segments. If there are *n* trajectory points for each divided segments satisfy the conditions: dis(*P*_center_, *p*) < *a* and |dir(*P*_1_, *P*_2_)-heading(*p*)| < *b* or |dir(*P*_2_, *P*_1_)-heading(*p*)| < *b*, then edge L was set as candidate true link. Where, *p* ∈ Tr, *P*_center_ is the center point of each segment, *P*_1_ and *P*_2_ are the start and end point of each segment, dis and dir are the function of the Euclidean distance and azimuth between two points, heading is the move angle of trajectory point. Here, considering both time cost and results precision, we recommend using the values *m* = 3, *n* = 5, *a* = 20 m, *b* = 30°.Criterion 2: if three sides of a triangle are identified as the candidate true links and at least one non-hypotenuse is not the hypotenuse of other triangles, the bevel edge of this triangle is defined as a false link.Criterion 3: if the candidate true link is a hanging edge, and the angle between this edge and other true links is less than 60°, it is a false link.Type II links identification: Since Delaunay triangulation network meets the maximum empty circle criteria, some true links between some intersections cannot be formed and identified, as the yellow line shows in B district of Figure 5b. Furthermore, this paper only constructs links based on road intersections; there will also have some missing links in the edge region, as the yellow line shows in A district of Figure 5b. Therefore, this paper integrates morphological methods [1] to optimize and supplement Type I links identified by Criterion 1, Criterion 2, and Criterion 3, which can help eliminate more false links by using other criteria and generate more precise road network. Preliminary identification results after optimization are shown in Figure 5c. The optimization steps are as follows:
Extracting missing road segments. A flat-head buffer with radius of *r*_2_ (by default, *r*_2_ = 50 m) was established based on candidate true links. Then, centerline extracted by morphological method can be classified missing road segments (red line) and matched lines (blue line), as shown in C district of Figure 5d.Repairing missing road segments. The short missing road segments were deleted first, and then we match the missing road segments to the corresponding intersections or end points of missing road segments by considering direction and distance, as shown in D district of Figure 5d.Generating Type II links. The end points of repaired missing road segments were connected to generate new true links.Type III links identification: the above processing focuses on dense areas, but there are still some true links, which are located in sparse areas and cannot be identified. Therefore, based on road structure features above mentioned, we propose some other criteria to identify Type III links from remaining links by removing false links. False link identification criteria are as follows:
Criterion 4: If two edges of a triangle are identified as true links, the third is defined as false link.Criterion 5: if the angle of one link and one true link at the common intersection is less than 60°, the link is the false link.Criterion 6: If one side of a triangle is true link and one side is false link, and if the last side is bevel edge, it must be false link.

#### 3.2.2. Adaptive Link Fitting

Based on the above process, dense GPS road links can be easily identified by Criteria 1, 2, and 3, and optimization, while sparse GPS road links can be judged by Criteria 4, 5, and 6. Different road links have different visual features and form different fitting methods.
Type I links fitting: the straight-line type I link identified by Criteria 1, 2, and 3, which coincides with road segments, can directly represent the centerlines of these road segments, as shown in Figure 5c.Type II links fitting: optimizing results can be used not only to eliminate false links, but also to effectively supplement road segment recognition results. Therefore, type II links identified by optimizing can also be fitted by optimizing results in turn, as shown in Figure 5d.Type III links fitting: type III links are located in sparse trajectory areas; it is difficult to identify pure true links. Therefore, in order to guarantee correctness of road generation results, we filtered type III links by judging whether there are sub-trajectory points between their end points (road intersections) and proposed a piece-wise fitting method to infer road centerlines for sparse GPS road segments.

For one trajectory *T*, if one sampling point *p_i_* is found within the buffer threshold *d*_1_ of intersection *I**_k_* and sampling point *p**_j_* is found within the buffer threshold *d*_2_ of another intersection *I**_j_*, the track segment (*p**_i_*, *p**_j_*) belongs to the section (*I**_k_*, *I**_j_*). Traverse all trajectory data, and calculate all track segments between *I**_k_* and *I**_j_*, sub-trajectory points can be obtained: (*I**_k_*, *I**_j_*)~{*p*_1_, *p*_2_, *p*_3_, …, *p**_n_*}.

The buffer threshold *d*_1_ and *d*_2_ are import parameters. If the thresholds are set too small, many sub-trajectories cannot be extracted. If they are set too large, road extraction results will include false road segments. Furthermore, the buffer threshold cannot be set to the same value for uneven distribution of trajectory data. According to observation, the scale of every intersection will not excess the distance of the shortest edge of the triangle with intersections as the common vertex. With intersections as the center of the circle and shortest edges as the radius, the scale range of each intersection can be well determined. Therefore, *d*_1_ and *d*_2_ can be set as the scale radius of intersections *I**_k_* and *I**_j_*, as shown in Figure 6a.

In order to eliminate sub-tracks that pass through the intersection *I**_k_* and then pass through other intersections for a long time to reach the intersection *I**_j_*, the following restrictions in distance, direction, and time were given. Take the link calculation between C_1_ and C_2_ in Figure 6b as an example, the limitations ensure that sub-trajectories between C_1_ and C_2_ do not pass through C_3_, C_4_.

Distance limitation: the length of GPS road segments between two intersections is generally not longer than its Euclidean distance between two intersections, which can be represented based on Equation (4).
(4)$$\mathrm{dis}(I_{k},p_{1})+\sum \mathrm{dis}(p_{i},p_{j})+\mathrm{dis}(p_{n},I_{j})\leq K_{1}D$$

Direction limitation: the heading direction of the vehicle does not change too much unless it turns at the intersections. Therefore, direction limitation (5, 6) is set to 60° according to the minimum intersection angle of two roads.
(5)$$|\mathrm{dir}(I_{1},\;I_{2})-\mathrm{heading}(p_{i})|\;\leq \;60{}^{\circ}$$
(6)$$|\mathrm{heading}(p_{1})-\mathrm{heading}(p_{n})|\;<\;60{}^{\circ}$$

Time limitation: for vehicle, the time it takes to travel directly from the starting point to the end point is generally less than the time it takes to reach the end point after passing through another intersection. Thus, we also set the time limitation, which can be represented based on Equation (7).
(7)$$(v_{1}+v_{n})(t_{n}-t_{1})/2\leq K_{2}D$$
where, *K*_1_ and *K*_2_ are the adjustment coefficient and generally set to 1.2 considering the curved road segments, *v_i_* is the speed of trajectory point *p_i_*, *t_i_* is the time-stamp of trajectory point *p_i_*, $\sum \mathrm{dis}(p_{i},p_{j})$ is the length of sub-trajectories passing through adjacent intersection points *I**_k_* and *I**_j_*, 1 < *I* < *j* < *n*, *D* represents the Euclidean distance between two intersection points.

The above sub-trajectory extraction results can be used, not only to determine whether type III links is true, but also to further fit the road segments of corresponding true links. Typically, road centerlines have higher point density and sub-trajectory points of Curved road segments often deviate some distance from the line between two intersections, as shown in Figure 7. Hence, we proposed piece-wise fitting method to create road segments. Considering low sampling frequency, these sub-trajectory points from *I**_k_* to *I**_j_* or *I**_j_* to *I**_k_* are all used for fitting road segments between *I**_k_* and *I**_j_*. More specifically, we divide the space into M parts successively from the beginning to the end point in the vertical direction of the line connecting two intersections. The corresponding sample Si is the max density point that located in the bin. Then we connect the start intersection point, the max density points, and the end intersection point into a line segment and adopt Douglas algorithm to simplify. Here, *M* = |*D*/*N*| and the density can be calculated based on Equation (8):(8)$$\rho_{i}=\sum_{j}\chi (d_{ij}-d_{c})$$
where χ(x)=1 if ($d_{ij}-d_{c}$) < 0 and χ(x)=0 otherwise, and the distance threshold in Douglas algorithm and the cutoff distance *d_c_* can be set as 20 m by default, *N* is the length of the bin (by default, *N* = 15 m).

## 4. Experiments and Analysis

### 4.1. Study Area and Data Sets

In order to reflect our methods’ performance, two old urban areas of Wuhan (Hankou District and Hongshan District) with different road structure layouts were used for experimental analysis, as shown in Figure 8a,b. The road network in Hankou District, which has a grid pattern distribution, is relatively regular. While the road network in Hongshan Square District, which is distributed in a circular radial pattern, exist much more complex situation. These two research areas not only have many old buildings, but also have many new buildings, which cause small distance between road intersections and a lot of noise in trajectories (Figure 8c,d). They are representative in the analysis and mining of road information. Moreover, [47] believes that when the collection period exceeds 7 days, the coverage of taxi data on the roads in Wuhan gradually stabilizes. Therefore, for the two research areas in Wuhan, we test our method based on the selecting 7-day taxi trajectory data from 29 May to 4 June, 2014. The sampling frequency of two data sets are mainly concentrated in 30–50 (s). Table 1 lists the basic statistics of these two data sets.

### 4.2. Results Evaluation and Analysis

Outlier points may affect the experimental results significantly. However, in order to extract more road intersections and road network, the process of road intersection extraction and road generation of our method only use the trajectories that have been removed the duplicate record points and some data whose heading is 0 and velocity is 0 or velocity is more than 100 km/h. For Hankou district, there are a total of 123 intersections, we initially extracted 217, within 50 m matching distance, the true value reaches 120. For Hongshan district, there are a total of 168 intersections, we initially extracted 289, within 50 m matching distance, the true value reaches 138. In Hankou District, some pseudo-intersections fall outside the road and need to be eliminated. To further ensure the accuracy of intersection extraction, false intersections connected by only two roads are also pruned based on road network extraction results. In this section, we compare our method with an incremental method of Ahmed [25], an intersection linking method of Karagiorgou [22] and a raster method of Davies [23]. Implementations of these three algorithms are provided by Ahmed [48].

#### 4.2.1. Visual Inspection

Obviously, our method obtained good results in both research regions (Figure 9), even for the Hongshan District, with more complex roads. The road segments near the intersections rarely present distortion and deformation. Road intersections or segments, which have quite sparse distribution of trajectories or locate close in space, are also identified. For the Wuhan dataset, with a lot of noise, the good results show that the method in this paper has anti-noise property, and can be applicable to the extraction of road networks in the old downtown areas.

However, for Ahmed’s method, there are many errors in the extracted road segments. Although Davies’ method has a good effect, it is difficult to apply to the low-density areas, and the road results are also distorted and have many burrs. The method of Karagiorgou, which is the intersection linking method, also cannot guarantee the correctness of road intersections, and generates many false links. Compared with our method, these three methods, which cannot extract more correct road intersections and segments, are not suitable for low frequency and high noisy trajectory data in old downtown areas.

#### 4.2.2. Quantitative Comparisons

We also made a quantitative comparative analysis of different road extraction methods, and calculated Precision, Recall, F-score from two aspects of road extraction results and intersection extraction results. For Wuhan dataset 1 and dataset 2, we downloaded OpenStreetMap, and then selected roads that were traversed by one and more trajectory as the ground-truth road networks. The indicator of Precision, Recall, F-score can be computed utilizing Equations (9)–(11).
(9)$$Precision=\frac{matched}{extracted}$$
(10)$$Recall=\frac{matched}{ground-truth}$$
(11)$$F-vaue=\frac{2*precision*recall}{precision+recall}$$

According to the method of Mariescu-Istodor [40], we converted the ground-truth road network and extraction road network into cells, and then calculated the difference of two sets. Our method always has the highest F-value in road extraction with the growing of grid resolution (Figure 10). The Karagiorgou method has relatively good recall, but the lowest precision, which means it generates many false road segments. Davies’ method has higher precision than our method at the grid resolution of 50 m, which was mainly caused by a large amount of burrs in extraction results of Davies. Most of these burrs are more than 40 m in length, as shown in Figure 9c,f. On the other hand, as the grid resolution increases, many identified adjacent roads are merged together, which increases the recognition rate of Davies’ method. Even though this method has a higher precision in 50 m grid resolution, it still has low recall and many missing road segments. This is consistent with the analysis results of visual inspection.

Although using different matching distances, our intersection extraction method also performs well, and has high Precision, Recall, and F-score, as shown in Figure 11. This suggests that the intersection location accuracy of other three method is not good; the road segments extracted near intersections are distorted. That is why our method has high precision in road extraction results. To some extent, this shows that the accuracy of intersections affects the results of road network extraction.

According to the observation from Figure 11, when the matching distance excesses 40 m, Precision, Recall, F-score of our road intersection results are stable. Correspondingly, all of the road evaluation indicators reach relatively large values at a grid resolution of 40 m, as shown in Figure 10. The distance between roads in the old downtown areas is small. If the grid resolution is set too large, the road results will be merged and affect the evaluation results. Here we set 40 m as the final comment searching scope. The comparison result of different methods is listed in Table 2.

## 5. Conclusions

The unique distribution characteristics of crowd-sourced trajectories and the complex and diverse road layout increase the difficulty of road network extraction in old downtown areas. Moreover, traditional road network extraction algorithms seldom consider structural characteristics of the road network in old downtown areas. Therefore, our objection was dedicated to generate a road network in old downtown areas, based on crowd-sourced trajectories. First, we focused on the intersections, and then constructed a road network based on the Delaunay triangulation network. During the process, the relative link identification criteria based on trajectory distribution and road structure features were proposed. We also fused the road extraction results of the morphology method to enhance the extraction integrity. Finally, a targeted link fitting strategy was proposed to generate a road network. The 7-day taxi trajectory data in Hankou and Hongshan district was used to test our method. It does not have complex pre-processing, can effectively avoid bad results caused by low quality data, and produce a relatively accurate and integral road network for old downtown areas. In sum, this method provides a promising solution for enriching and updating road networks for old downtown areas, and can be applied in navigable road network construction, intelligent transportation systems, and city planning.

However, it still has some limitations. Due to the inherent problems of high noise, low frequency, and low precision for experimental data, some road segments with extremely sparse trajectories cannot be extracted, and some road segments with high noise are incorrectly identified. In fact, these two defects listed above are essentially due to low data quality and are difficult to be solved through single data source, which would orient a future study, to fuse other sourced data, such as pedestrian trajectories and remote sensing images, to further supplement true road segments and eliminate false road segments.

## Figures and Tables

**Figure 1 sensors-21-00235-f001:**
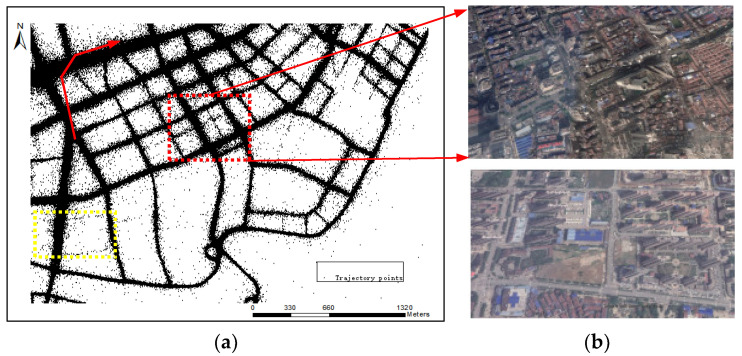
The road network example. (**a**) Trajectory distribution in Hankou old downtown areas (red line is a single trajectory); (**b**) remote sensing images of Hankou old downtown areas and new downtown areas in Jiangxia.

**Figure 2 sensors-21-00235-f002:**
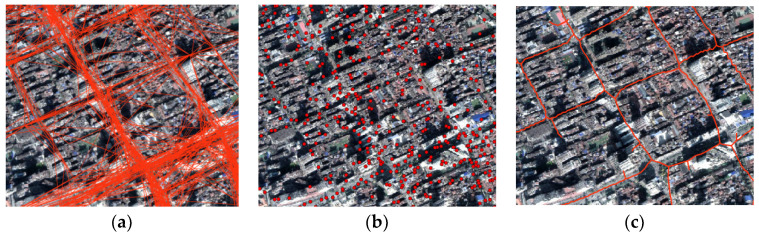
Extraction results of existing methods for old downtown areas. (**a**) The method of Cao; (**b**) the method of Edelkamp; (**c**) the method of Davies.

**Figure 3 sensors-21-00235-f003:**
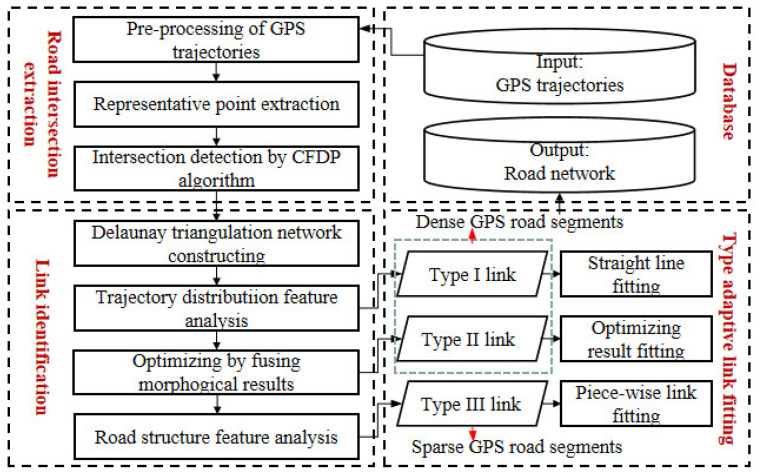
Workflow of road network construction for old downtown areas.

**Figure 4 sensors-21-00235-f004:**
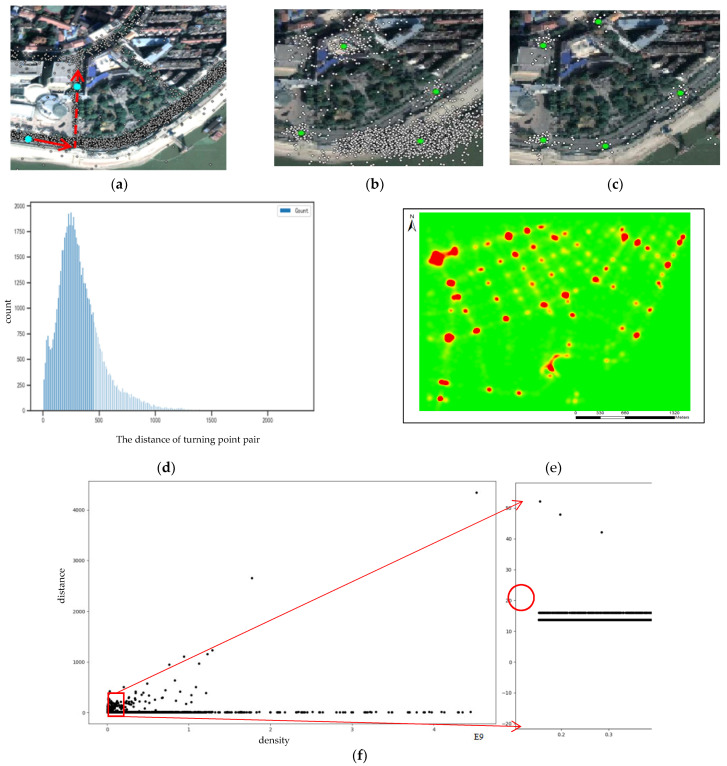
Road intersections extraction. (**a**) Turning point vectors; (**b**) converging points; (**c**) extraction results of distance limited; (**d**) distance statistics; (**e**) Kernel density; (**f**)The decision graph of $\delta_{qi}$ and $\rho_{i}$ (the right picture is part of the amplified result).

**Figure 5 sensors-21-00235-f005:**
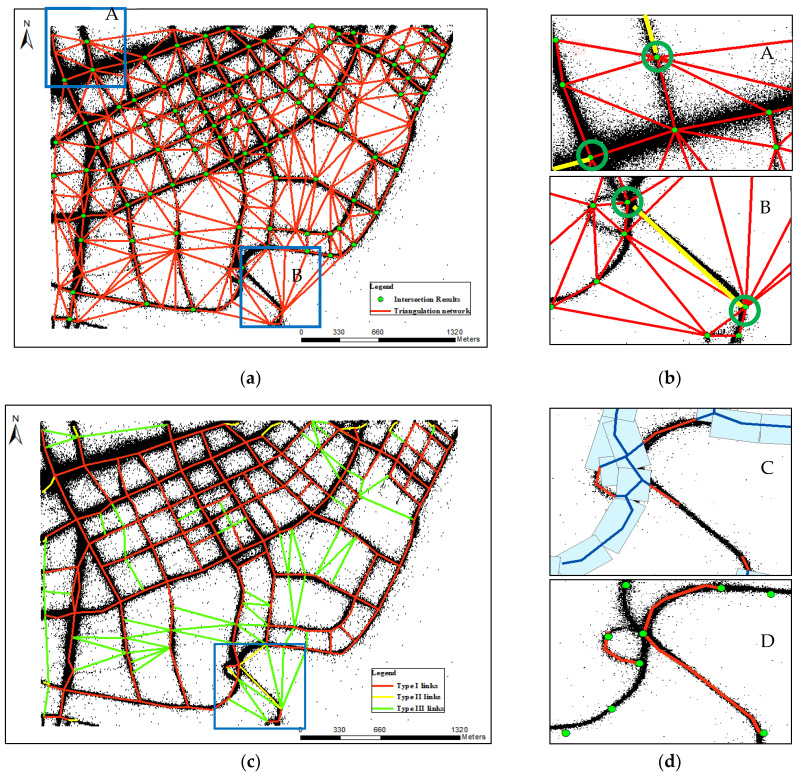
Links identification. (**a**) Initial links identification network; (**b**) missing true links due to the largest empty circle criterion and being located in the edge area; (**c**) preliminary identification results; (**d**) optimizing example.

**Figure 6 sensors-21-00235-f006:**
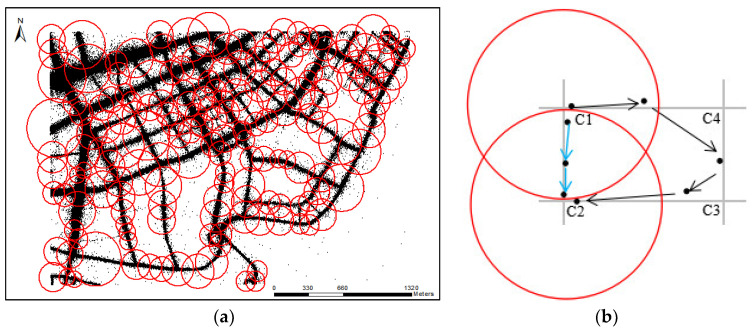
Sub-trajectory extraction. (**a**) Intersection scale estimation; (**b**) Sub-trajectory limitation.

**Figure 7 sensors-21-00235-f007:**
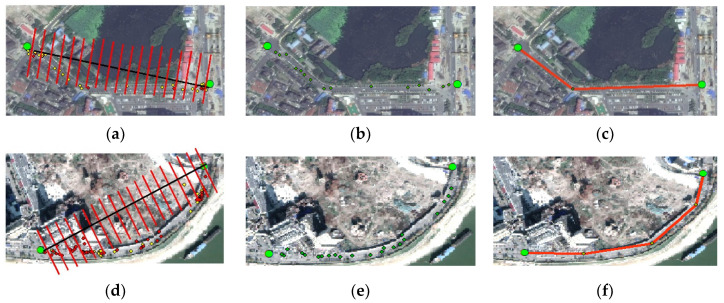
Illustration of different segments fitting. (**a**,**d**) Space dividing (**b**,**e**) results extraction based on density sampling (**c**,**f**) segment creating based on Douglas algorithm.

**Figure 8 sensors-21-00235-f008:**
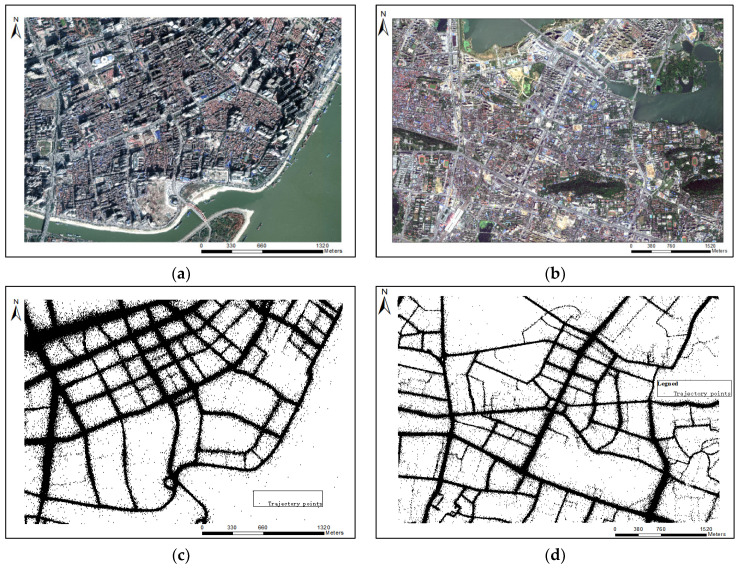
Trajectory datasets: (**a**) remote sensing image for Wuhan1 in Hankou District; (**b**) remote sensing image for Wuhan2 in Hongshan District; (**c**) trajectory dataset for Wuhan1 in Hankou District; (**d**) trajectory dataset for Wuhan2 in Hongshan District.

**Figure 9 sensors-21-00235-f009:**
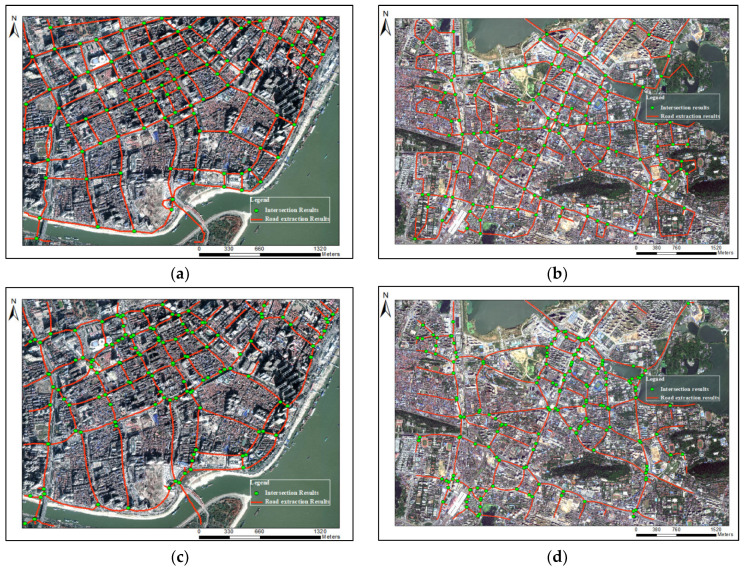
Road network and intersection extraction results. (**a**) Results of our method for Wuhan dataset 1; (**b**) results of our method for Wuhan dataset 2; (**c**) results of Davies’ method for Wuhan dataset 1; (**d**) results of Davies’ method for Wuhan dataset 2; (**e**) results of Ahmed’s method for Wuhan dataset 1; (**f**) results of Ahmed’s method for Wuhan dataset 2; (**g**) results of Karagiorgou’s method for Wuhan dataset1; (**h**) results of Karagiorgou’s method for Wuhan dataset 1.

**Figure 10 sensors-21-00235-f010:**
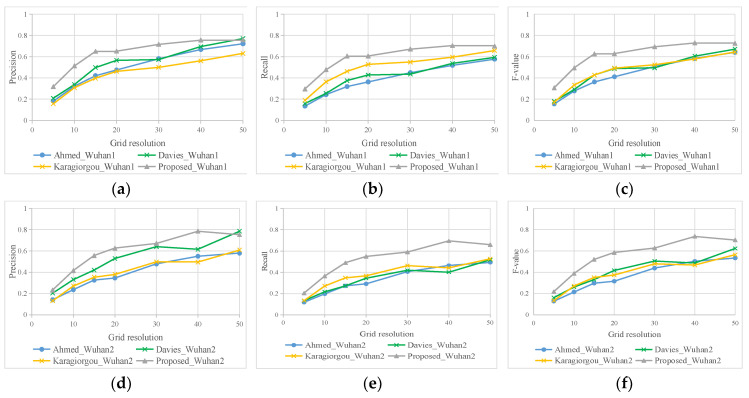
Quantitative comparisons of road extraction. (**a**–**c**) Wuhan dataset 1; (**d**–**f**) Wuhan dataset 2.

**Figure 11 sensors-21-00235-f011:**
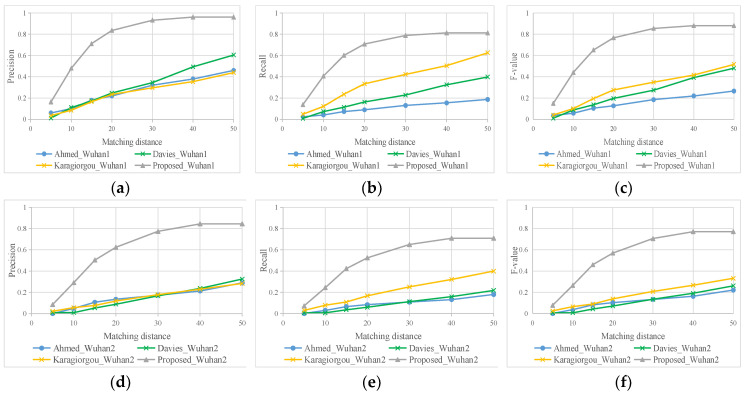
Quantitative comparisons of intersection extraction. (**a**–**c**) Wuhan dataset 1; (**d**–**f**) Wuhan dataset 2.

**Table 1 sensors-21-00235-t001:** Statistics of these two data sets.

Data Set	Trajectory Points	Average Sampling Rate(s)	Area (km^2^)	Average Speed (km/h)
Data set 1	800,868	>45	4.2 × 2.8	31.6
Data set 2	1,343,409	>45	5.7 × 3.9	33.2

**Table 2 sensors-21-00235-t002:** Comparison of experimental results.

Dataset	Method	Intersection Extraction			Road Segment Extraction		
		Precision	Recall	F-Value	Precision	Recall	F-Value
Dataset 1	Proposed	96.2%	81.3%	88.1%	75.6%	70.4%	72.9%
	Davies	49.4%	32.5%	39.2%	69.5%	53.7%	60.6%
	Ahmed	38.0%	15.4%	22.0%	66.8%	51.9%	58.4%
	Karagiorgou	35.4%	50.4%	41.6%	56.1%	59.6%	57.8%
Dataset 2	Proposed	84.4%	70.8%	77.0%	78.5%	69.4%	73.7%
	Davies	23.7%	15.9%	19.0%	61.5%	40.1%	48.6%
	Ahmed	21.2%	13.1%	16.2%	55.0%	46.3%	50.3%
	Karagiorgou	22.8%	32.1%	26.7%	49.5%	44.2%	46.7%

## Data Availability

Data are not publicly available.

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
