# Peer review of "Generating Road Networks for Old Downtown Areas Based on Crowd-Sourced Vehicle Trajectories"

_sensors, 2021, doi:10.3390/s21010235_

Round 1
Reviewer 1 Report
Dear Authors,
I have enjoyed reading your manuscript. It is very interesting. Below please find some suggestions:
- In section 3. Road network generation Method. You suggest different methods for different fitting. Please, add clear justification of this solution. There are also distinguished three link types: Tyoe 1, 2 and 3. I did not find any explanation what it means. Please, add such explanation.
- All scale bars in all figures need cartographic improvement. The width of the scale bar should be round value, easy dividable by 10, bo decimal numers should appear on the scale bar.
- In section 4. Experiments and analysis. You have considered only Wuhan vincity. There should be some justification of this. How is it representative? And what is the difference between two considered districts?
- In the subsection 4.2.2 Quantitative Comparisons. Please consider adding a table presenting Precision, Recall and F-values.
- Figure 10 and 11 - the font on the legends needs enlargement. Now it is difficult to read.
Best regards
Author Response
Dear professor, Please see the attachment, thank you.

Reviewer 2 Report
Thank you for your contribution on "generating road networks for old downtown based on crowd.sourced vehicle trajectories", which for my point of view is an excellent paper.
The problem is clearly described and the problems of related work are extensively shown, especially for the study areas. The development of methods and the three step approach is explained in detail in section 3. The advantages as well disadvantages and shortcomings are highlighted for the main approach, which makes this method very objective. It fits into the actual discussion and developments in this working area.
Although the reviewer has never been in the study area, the images of the old town areas show the difficult and problematic constellation for generating road networks. If the method works in these areas, then it will work in many others too. The study areas are nicely choosen.
Gratulation to and thanks for this exciting contribution.
Author Response
Dear professor, thanks for your encouragement and recognition. It is very important to me. It improves my confidence in future research. I will try my best to do follow-up research and contribute my own exploration in this field.
Reviewer 3 Report
The current paper presents an intersection-first approach to generate road network based on low-quality crowd-sourced vehicle trajectories. It is an interesting approach that is well-presented. I suggest accepting the manuscript after major revisions.
Line 144: How much did you limit the distance?
Line 148: Which were the rules?
Line 172: How did you define the threshold K? Did you use KDE as a map-matching process?
Line 192: How much was the percentage of pseudo intersections?
Line 201: how did you come up with the division with the number 250?
Line 341: Despite the frequency of the sampling data, which is very good. How many GPS points in total did you use for the study?
Figure 10a: Did you test what happens for grid resolution higher than 50 concerning the Davies methodology, which seems to have higher precision for this resolution?
Conclusions:
- What is the minimum data volume that is required to identify a road segment?
- Were there any cases that you couldn’t detect a road segment because of the lack of data?
- For who do this methodology would be beneficial?
- How much computing time do you need for this process?
- On what computer did you test it? Does your approach require specific computer characteristics to run?
Author Response
Dear professor, please see the attachment, thank you.

Round 2
Reviewer 3 Report
The authors have improved the quality of the paper significantly. I suggest accepting in the current form.